# Non-Feeding Transmission Modes of the Tomato Yellow Leaf Curl Virus by the Whitefly *Bemisia tabaci* Do Not Contribute to Reoccurring Leaf Curl Outbreaks in Tomato

**DOI:** 10.3390/insects15100760

**Published:** 2024-09-30

**Authors:** Wendy G. Marchant, Judith K. Brown, Saurabh Gautam, Saptarshi Ghosh, Alvin M. Simmons, Rajagopalbabu Srinivasan

**Affiliations:** 1Department of Entomology, University of Georgia, Griffin, GA 30223, USA; wgmarch@att.net (W.G.M.); saurabh@apcd.ca.gov (S.G.); sunnysaptarshi@gmail.com (S.G.); 2School of Plant Sciences, University of Arizona, Tucson, AZ 85721, USA; jbrown@ag.arizona.edu; 3Agriculture Research Service, United States Department of Agriculture, Charleston, SC 29414, USA; alvin.simmons@usda.gov

**Keywords:** *begomovirus*, sexual transmission, transovarial transmission, plant transmission

## Abstract

**Simple Summary:**

Tomato yellow leaf curl virus (TYLCV) poses a serious constraint to tomato production in the Southeastern United States. Since its introduction, it has become endemic in states such as Georgia and Florida despite tomato-free periods. This study examined whether the endemic status of TYLCV is aided by the *Bemisia tabaci* B cryptic species’ vertical and/or horizontal non-feeding modes of TYLCV (the Georgia variant of the Israel strain) transmission. The potential for ‘non-feeding’ transmission modes has been examined previously, with inconsistent results. The objective of this study was to determine the frequency of TYLCV transovarial and/or sexual (mating) transmission and the subsequent transmissibility of the virus to virus-free plants. The study utilized a series of virus transmission assays and detection and quantitation tools. Our results indicate that non-feeding transmission modes are a rare phenomenon in this pathosystem, and their significance is irrelevant for the observed endemic status of TYLCV in the southeastern United States.

**Abstract:**

Tomato yellow leaf curl virus (TYLCV) causes significant yield loss in tomato production in the southeastern United States and elsewhere. TYLCV is transmitted by the whitefly *Bemisia tabaci* cryptic species in a persistent, circulative, and non-propagative manner. Unexpectedly, transovarial and sexual transmission of TYLCV has been reported for one strain from Israel. In this study, the potential contribution of the *B. tabaci* B cryptic species transovarial and sexual transmission of TYLCV (Israel strain, Georgia variant, Georgia, USA) to reoccurring outbreaks was investigated by conducting whitefly-TYLCV transmission assays and virus DNA detection using end point PCR, DNA quantitation via real-time PCR, and virion detection by immunocapture PCR. TYLCV DNA was detectable in four, two, and two percent of first-generation fourth-instar nymphs, first-generation adults, and second-generation adults, respectively, following transovarial acquisition. Post-mating between viruliferous counterparts, the virus’s DNA was detected in four percent of males and undetectable in females. The accumulation of TYLCV DNA in whiteflies from the transovarial and/or sexual experiments was substantially lower (100 to 1000-fold) compared with whitefly adults allowed a 48-hr acquisition-access period on plants infected with TYLCV. Despite the detection of TYLCV DNA in whiteflies from the transovarial and/or mating experiments, the virions were undetectable by immunocapture PCR—a technique specifically designed to detect virions. Furthermore, tomato test plants exposed to whitefly adults that presumably acquired TYLCV transovarially or through mating remained free of detectable TYLCV DNA. Collectively, the extremely low levels of TYLCV DNA and complete absence of virions detected in whiteflies and the inability of the *B. tabaci* cryptic species B to transmit TYLCV to test tomato plants following transovarial and mating acquisition indicate that neither transovarial nor sexual transmission of TYLCV are probable or epidemiologically relevant for TYLCV persistence in this pathosystem.

## 1. Introduction

Begomoviruses are whitefly-transmitted plant pathogens that cause yield-limiting diseases in diverse agricultural crops [1]. The genus *Begomovirus* is the largest among nine genera classified in the family *Geminiviridae*. Begomoviruses infect dicotyledonous plants and are transmitted by the whitefly *Bemisia tabaci* (Gennadius) cryptic species complex (Hemiptera; Aleyrodidae) [2,3]. The begomovirus genome comprises either one or two circular ssDNA components that are approximately 2.6 or 5.2 kilo bases (kb) in size, respectively. Each ssDNA component is encapsidated in a twinned, icosahedral, or ‘geminate’ particle [4,5,6].

The monopartite tomato yellow leaf curl virus (species *Begomovirus coheni*) originated in the Middle East and belongs to the genus *Begomovirus* [7,8]. Infection by tomato yellow leaf curl virus (TYLCV; Israel strain) can cause 100% infection and substantial yield loss in commercial tomato fields and greenhouses in the United States and elsewhere [9,10]. Symptoms of TYLCV infection are characterized by foliar chlorosis, upward or downward curling of leaves, stunting of the plant, and reduced fruit set [9]. TYLCV is transmitted by *B. tabaci*, considered by most to be a cryptic species complex [11,12,13]. The *B. tabaci* cryptic species complex comprises six or more morphologically indistinguishable but biologically distinct phylogeographic groups/clades based on phylogenetic analysis of the mitochondrial cytochrome oxidase subunit I (COI) gene, nuclear orthologs (>2000) of the whitefly genome, and microsatellite markers [2,11,12]. Based on phylogenomic analyses of nuclear orthologs, mitotypes B and Q, endemic to the North Africa/Middle East and Mediterranean regions, respectively, have been resolved as the NAFME and MED cryptic species (previously, NAFME was known as the B biotype/mitotype or Middle East-Asia Minor 1 (MEAM1) and MED was known as the Q biotype/mitotype or Mediterranean (MED)). In this study, the NAFME and MED cryptic species (nuclear orthologs) or corresponding mitotypes (mtCOI), are referred to as the B and Q cryptic species, respectively. It appears that several haplotypes of the B and Q cryptic species have invaded non-endemic locales [11,12,13], and they displaced relatively benign native cryptic species [11,14,15,16]. The combined invasiveness of the B and Q cryptic species and their propensity to colonize tomato crops and transmit TYLCV have facilitated the rapid spread of TYLCV (Israel strain) throughout the world in recent years [16,17].

The TYLCV, like all begomoviruses, is transmitted in a persistent, circulative, and non-propagative mode by *B. tabaci* cryptic species [18,19,20,21,22,23]. However, three studies have reported evidence for low levels of virus transcripts and/or possible replication of TYLCV in *B. tabaci* [24,25,26]. Begomovirus particles (virions) are ingested by whiteflies during phloem-feeding on a virus-infected plant. Following ingestion, the virions are taken up through the stylet and enter the alimentary canal, where they are translocated across the midgut barrier and/or filter chamber into the whitefly hemolymph, from where they are translocated into the salivary glands by (putative) receptor-mediated endocytosis [27,28]. Once internalized in the primary salivary glands, the virions are inoculated to the host plant, or ‘transmitted’, in the saliva during feeding. The minimum and maximum latent period required from ingestion to entry into the salivary glands is about eight and twenty-four hours, respectively, and once a whitefly becomes ‘viruliferous’ or capable of transmission, it remains so for its lifetime [18,19,20,21].

Studies carried out in several laboratories have examined the transovarial and sexual transmission of TYLCV by the B cryptic species [29,30,31,32,33,34,35]. These non-feeding virus modes of transmission are rather unusual among the genus *Begomovirus*. Two studies have reported that TYLCV DNA was detectable in eggs, nymphs, and adults of the F1 generation, at 81–92, 37–68, and 57–77%, respectively, and in second-generation (F2) offspring eggs, nymphs, and adults at frequencies ranging from 3–79% [31,34]. Ghanim et al. (1998) and Wei et al. (2017) also documented plant transmission levels following transovarial acquisition of up to 80% [31,34]. At the same time, conflicting results have demonstrated that transovarial transmission was of no epidemiological relevance for either TYLCV or another monopartite begomovirus species, the tomato yellow leaf curl Sardinia virus (TYLCSV), or the bipartite tomato yellow leaf curl Thailand virus (TYLCTHV) [22,29,30,31,32,33]. Only two to three percent of adult Q cryptic species (F1 offspring) were found to be positive for TYLCV DNA when examined using PCR amplification [32]. Further evidence for lack of transmission by non-feeding mechanisms has been provided by two independent transovarial transmission studies in B cryptic species adults (F1 offspring) [29,33]. 

In one study involving sexual transmission, TYLCV DNA was detected either mono- or bi-directionally between mated pairs of both the B and Q cryptic species [15]. With respect to the B cryptic species, TYLCV DNA was detected in the male-to-female direction alone at a 10% frequency, while for the Q cryptic species, virus DNA was detected in female-to-male and male-to-female crosses with frequencies of 50 and 74%, respectively [15]. An earlier study from Israel demonstrated that mating transmission ranged from 28 to 56%, and when whiteflies that presumably acquired the virus during copulation (mating) were clip-caged to non-infected tomato plants, up to 35% of the plants became infected with TYLCV [35]. This was the only report of putative sexual transmission of TYLCV that resulted in symptomatic infection in plants. With so few studies that addressed mating transmission, it was not clear how prevalent the mating transmission of TYLCV is among *B. tabaci* populations. With mild winters, whitefly feeding and reproduction slow but are not abated altogether unless temperatures fall below freezing and the plant host also dies. If virus transmission indeed occurs widely through mating and the transovarial transfer of virions to offspring, TYLCV spread within the *B. tabaci* populations would be expected to result in increased inoculation potential.

Feeding-based virus transmission assays have indicated that differences in transmission efficiencies are influenced by TYLCV variants and perhaps by *B. tabaci* haplotypes [36]. The objective of this study was to determine the prevalence of transovarial and mating transmission of TYLCV (the Georgia variant of the TYLCV strain) and its subsequent transmissibility to non-infected plants via the representative B cryptic species haplotype in the southeastern state of Georgia within the United States. The mechanisms by which TYLCV over-winter and over-summer during tomato-free periods are not yet entirely known. Evaluations of crop hosts, ornamentals, and weed hosts did not result in TYLCV infection and/or back transmission to tomatoes in southeastern states (Florida and Georgia) in many instances [37,38]. Despite the relatively narrow TYLCV host range, TYLCV incidences have become chronic in Georgia even with tomato-free seasons. This led to the speculation that non-feeding transmission of TYLCV could aid in sustaining the inoculum within whiteflies across seasons. Until now, the two non-feeding modes of transmission by this specific B cryptic species-TYLCV complex have not been considered epidemiologically relevant to initial or reoccurring outbreaks. 

## 2. Materials and Methods

### 2.1. Maintenance of the Whitefly Colony and TYLCV Variant

Whiteflies identified as the B cryptic species based on phylogenetic analysis of the COI sequence using the Frohlich et al. (1999) [39] protocol were collected during 2011 from infested cotton plants in Tifton, Georgia, and maintained on cotton plants (approximately 8–10 leaf stage; 15–20 cm in height) in a whitefly-proof cage within an otherwise insect-free greenhouse at 25–30 °C and 14 h L:10 h D photoperiod. The TYLCV variant was collected from a symptomatic tomato plant in Tifton (Tift County, GA, USA) in 2015, and its identity was confirmed based on DNA sequencing of the cloned virus genome (Accession no. KY965880). The TYLCV variant from Georgia was maintained in tomato plants (6 leaf stage) by inoculation with viruliferous whitefly adults, and subsequent serial transfer (approximately every 4–6 weeks) to tomato plants of the cultivar Florida 47 (Seminis Vegetable Seeds, St. Louis, MO, USA). The TYLCV infected source plants were maintained under the temperature and photoperiod conditions described above in a greenhouse separate from the one containing the whitefly colony. 

### 2.2. Detection of TYLCV DNA in Offspring of Viruliferous Whiteflies 

Non-viruliferous whiteflies reared on TYLCV-free cotton plants were mechanically aspirated and used to generate viruliferous whiteflies. Viruliferous whiteflies were obtained by providing newly emerged adults (~3 days old; achieved by rearing in cohorts) a three-day acquisition-access period (AAP) on TYLCV-infected tomato plants using clip cages. For the transovarial evaluation experiment, viruliferous female whiteflies (less than one week old) were clip caged (up to 10 per clip cage) and allowed to oviposit on cotton, a non-host of TYLCV, for one week, after which the female whiteflies and clip cages were removed from the cotton plants and tested for the presence of TYLCV DNA using PCR amplification. Briefly, individual whiteflies were surface sterilized to remove honeydew or other possible contaminants, using a series of five 500 µL washes: 70% ethanol, water, 1% bleach, water, and water [21,40]. The final rinsate was collected and subjected to PCR amplification for TYLCV detection as described. Surface sterilization was carried out because honeydew might contaminate the exterior surfaces and potentially lead to false positives [21]. DNA was isolated and purified from whiteflies using Instagene Matrix following the manufacturer’s recommendations (Biorad, Hercules, CA, USA). 

The primers used were C2-1201(5′-CATGATCCACTGCTCTGATTACA-3′) and C2-1800V2 (5′-TCATTGATGACGTAGACCCG-3′), with an expected size product of 695 bp. The PCR reaction was carried out in 10 μL vol with 5 μL GoTaq^®^ Green Master Mix (Promega Corporation, Madison, WI, USA), 2 μL water, 0.5 μL of each primer at a concentration of 10 μM, and 2 μL of total DNA. The PCR parameters included an initial denaturation step at 94 °C for 2 min followed by 30 cycles of 94 °C for 30 s, 52 °C for 30 s, 72 °C for 1 min, and a final extension at 72 °C for 5 min. 

Cotton plants with eggs (post oviposition) were caged in whitefly-proof cages and placed in a growth chamber at 25–30 °C with a 14 h L:10 h D photoperiod to avoid the introduction of outside whiteflies. Fifty fourth-instar nymphs (F1) were removed from the leaf surface and individually collected into 1.5 mL microcentrifuge tubes. After surface sterilization, total DNA was extracted as indicated above. The DNA extracts of individual fourth-instar nymphs were subjected to PCR amplification to determine whether TYLCV DNA was transferred from mother to offspring. In addition, this experiment was repeated to collect (50) adult offspring (F1), and of the next (F2 or second) generation (50) adult offspring reared on cotton plants. Whiteflies allowed a 48-h AAP on TYLCV-infected tomato source plants were used as the PCR-positive control. Two replicated experiments were carried out for nymphs, adult offspring, the second-generation adult offspring (n = 100 for each stage).

### 2.3. Detection of TYLCV DNA in Mated Whiteflies

The sex of the adult whiteflies was determined by immobilizing the whiteflies on chilled packed blue ice and examining them under a dissecting microscope at 10× magnification. Individual viruliferous male and female whiteflies less than seven days old were transferred to a cotton leaf and confined in a clip cage with a non-viruliferous whitefly, less than three days old, of the opposite sex, and were allowed a mating access period (MAP) of 48 h. The whiteflies were then collected, surface sterilized (as described above), and assayed for the presence of TYLCV DNA by PCR. The whitefly DNA was isolated and purified as described, using Instagene Matrix. The ‘initially-viruliferous’ whiteflies were tested with PCR amplification to determine whether they harbored TYLCV DNA. Also, the initially non-viruliferous whiteflies were assayed by PCR to determine whether TYLCV DNA was present in them following mating. Whiteflies exposed to TYLCV-infected tomato source plants for a 48 h AAP were assayed by PCR as positive controls. Each experiment utilized 25 pairs of TYLCV-positive males and TYLCV negative females, and 25 pairs with the respective inverse combination, and was carried out two times (n = 50 for each mating combination).

### 2.4. Quantitation of TYLCV DNA in Positive Whitefly Samples

Whiteflies identified as PCR-positive for TYLCV DNA detection in experiments designed to test for transovarial and mating transfer were subjected to real-time PCR to quantitate the amount of TYLCV DNA the whiteflies had acquired. Whiteflies that had ingested TYLCV DNA by phloem-feeding on symptomatic TYLCV source plants were used as the positive control. The primers used to quantitate the C2 gene fragment of TYLCV were C2F (5′-GCAGTGATGAGTTCCCCTGT-3′) and C2R (5′-CCAATAAGGCGTAAGCGTGT-3′). The real-time PCR amplification was performed in a 25 μL reaction with 12.5 μL of GoTaq^®^ qPCR Master Mix (Promega Corporation, Madison, WI, USA), 6.5 μL of water, 0.5 μL of each primer at 10 μM concentration, and 5 μL of DNA template. The amplification parameters included an initial denaturation at 95 °C for 2 min followed by 40 cycles of 95 °C for 15 s and 60° C for 1 min, followed by a melting curve analysis. The PCR results were normalized using the whitefly β-actin gene as the internal whitefly baseline gene, which was amplified with primers for whitefly β-actin F (5′-TCTTCCAGCCATCCTTCTTG-3′) and whitefly β-actin R (5′-CGGTGATTTCCTTCTGCATT-3′) [24]. Relative quantitation was calculated for the TYLCV and whitefly β-actin C_T_ values using the Pfaffl equation [41]. 

### 2.5. Transovarial and Sexual Acquisition of TYLCV and Subsequent Transmissibility to Plants

Viruliferous female whitefly adults less than one week old were allowed to oviposit on cotton, a TYLCV non-host, for one week using clip cages as explained above. After one week, a sub-cohort (n = ~10) of female whitefly adults were removed and tested by PCR amplification as described above to verify the virus’s presence or absence. Cotton plants that had been infested by the TYLCV-positive cohort were placed in insect/whitefly-proof cages and maintained in an insect-free growth chamber. Adult offspring (F1) were collected post-eclosion from cotton plants, and 50 adult offspring were transferred to a clip cage on a tomato test plant and allowed a 48 h IAP. After the IAP, the whiteflies were removed along with the clip cage. The tomato plants were placed in whitefly-proof cages for four weeks in a growth chamber and observed for the development of symptoms characteristic of TYLCV infection. After four weeks, newly developing tomato leaves were collected and subjected to total DNA isolation followed by PCR amplification to detect the presence of TYLCV DNA. Two replicated experiments were carried out with ten plants each. 

To test for the transfer of TYLCV during mating (sexual transmission) and the possible transmission during feeding on tomato test seedlings, 50 male and female viruliferous whiteflies were transferred to cotton leaves confined to a clip cage with fifty non-viruliferous whiteflies of the opposite sex aged less than 3 days for a 48 h MAP. Twenty-six whiteflies (initially non-viruliferous) from cotton clip cages were transferred to virus-free tomato test plants (26/plant) and given a 48 h IAP. The tomato plants were held in whitefly-proof cages and maintained in an insect-free growth chamber, with conditions as described above. The tomato plants were maintained for four weeks to allow the development of an infection. The total DNA was isolated and purified from newly developed tomato leaves and subjected to PCR amplification for TYLCV DNA detection. Six plants were used per experiment, and the experiment was conducted twice for positive male/negative female combination, and the reciprocal combination.

### 2.6. Detection of TYLCV Virions in Whiteflies from Transovarial and Sexual Transfer Experiments

Immunocapture PCR was performed to determine whether virions could be detected within the whitefly body following transovarial and/or sexual acquisition, which would be indicative of virus replication and virion encapsidation within the whitefly and/or of the direct transfer of virions, respectively.

Adult offspring given the opportunity to acquire TYLCV transovarially (F1 and F2) were collected from cotton plants and subjected to immunocapture PCR amplification to detect the presence of TYLCV virions. The experiment consisted of homogenates from 50 offspring obtained according to a previously described method [21]. Immunocapture PCR was performed on 50 whitefly offspring homogenates (individually) using the polyclonal antibody raised against the TYLCV coat protein (Bioreba, Ebringen, Germany) [35]. The PCR tubes (200 μL) were coated with TYLCV antiserum at a 1:1000 dilution in 100 μL of coating buffer. The coated tubes were incubated for 3 h at 37 °C and washed with 100 μL of washing buffer three times. The washed tubes were then incubated with 100 μL of individual whitefly homogenates for 18 h at 4 °C. Following the incubation period, PCR amplification was conducted as described above for transovarial and mating experiments using the primer pairs C2-1201 and C2-1800V2. Whiteflies allowed a 48 h AAP on TYLCV infected tomato plants were used as the PCR-positive control. This experiment was conducted twice.

Whitefly mating pairs were assembled as described above, with twenty-five mating pairs of positive male/negative female and with the reciprocal combination (n = 25 for each mating combination). Whiteflies allowed a 48 h MAP were surface sterilized as described earlier. The immunocapture PCR amplification assay was performed on homogenates of the mated adult whiteflies (individually). This experiment was conducted twice. 

### 2.7. Comparison of TYLCV Georgia Variant Coat Protein Sequences with Other TYLCV Variants

Three coat protein sequences from TYLCV Georgia variants were compared with 12 other TYLCV sequences representative of different parts of the world (Table 1). The Georgia TYLCV coat protein sequences used in this study were all obtained from whole genome sequencing of TYLCV from tomato leaf samples collected in Georgia in 2015 and 2016. The sequences were aligned using Geneious Pro V.8.1.9 [42] using the global alignment with free end gaps option with a gap open penalty of 12, gap extension penalty of 3, and refinement iteration of 2. 

## 3. Results

### 3.1. Detection of TYLCV DNA in Offspring of Viruliferous Whiteflies

TYLCV DNA was detected by PCR amplification in the first (F1)-generation nymphs and first (F1)- and second (F2)-generation offspring adults originating from viruliferous adults (mothers) and developed post-eclosion on cotton—a non-host for TYLCV (Figure 1). TYLCV DNA was detected only in four percent of fourth-instar nymphs, and in two percent of both the first- and second-generation adult offspring (Table 2).

### 3.2. Detection of TYLCV DNA in Mated Whiteflies 

TYLCV DNA was detected in initially non-viruliferous adults that mated with viruliferous members of the opposite sex (Figure 1). In this experiment, TYLCV DNA was detected in four percent of the initially non-viruliferous males that mated with viruliferous females (Table 3). However, among the initially non-viruliferous females that mated with viruliferous males, none tested positive for the presence of TYLCV DNA (Table 3).

### 3.3. Quantitation of TYLCV DNA in Positive Whitefly Samples

TYLCV DNA was quantitated in whiteflies that tested positive from the transovarial and sexual transfer experiments. The levels of TYLCV DNA accumulation in whitefly offspring (F1) from the transovarial transfer experiment steadily declined with molting, and in the subsequent (F2) generation (Figure 2). The first-generation fourth-instar nymphs had the highest concentration of TYLCV DNA, followed by first-generation adult whiteflies, and the least in second-generation adult whiteflies (Figure 2). Whiteflies that acquired TYLCV during the transovarial transfer experiment harbored lower amounts of virus DNA compared with the positive experimental control, which consisted of whiteflies that were exposed to TYLCV by phloem-feeding on virus-infected tomato plants (Figure 2). Similarly, male whiteflies that acquired TYLCV through mating also had reduced amounts of virus DNA compared to the positive control whiteflies that were exposed to TYLCV by phloem-feeding on virus-infected tomato plants (Figure 2). The amount of TYLCV DNA detectable in the whiteflies following transovarial and/or sexual acquisition were consistently one to three orders of magnitude less than when TYLCV acquisition was demonstrated following feeding on TYLCV-infected plants (Figure 2).

### 3.4. Transovarial and Sexual Acquisition of TYLCV and Subsequent Transmissibility to Plants

All the parent (F0) whiteflies of the sub-cohort that acquired TYLCV via phloem-feeding tested positive for TYLCV. The first-generation adult offspring from viruliferous mothers, when provided with a 48 h IAP on tomato test plants, did not cause the development of symptoms characteristic of TYLCV infection, nor was TYLCV DNA detected in the ‘inoculated’ test plants based on the results of PCR amplification (Table 4). Thus, transmission to tomato plants was not demonstrated following the putative transovarial acquisition of TYLCV. Similarly, when initially non-viruliferous whiteflies that mated with viruliferous members of the opposite sex were allowed a 48 h IAP on non-infected tomato plants, there was no development of symptoms characteristic of TYLCV, and the plants were negative when assayed by PCR amplification (Table 4). In no experiment was TYLCV shown to be transmissible to tomato plants following whitefly copulation. 

### 3.5. Transovarial and Sexual Transfer of TYLCV Virions

No TYLCV virions were detected by the immunocapture PCR assay in the first-generation (F1) of whitefly offspring from viruliferous females (mothers) (Table 5). Virions were also not detected using immunocapture PCR in any of the initially non-viruliferous whiteflies that mated with TYLCV viruliferous whiteflies (Table 5). By comparison, TYLCV virions were consistently detectable by immunocapture PCR amplification in whiteflies that were allowed a 48 h AAP on the TYLCV-infected tomato source plants (Figure 3). 

### 3.6. Comparison of TYLCV Coat Protein Sequences of Multiple Variants

An alignment of the amino acid sequences for the coat protein of the TYLCV isolated from the Georgia and 12 other representative TYLCV variants revealed that for the most part, the amino acid sequences varied by single amino acid substitutions (19 in total), with two exceptions. The China variant (Accession no. KM 435335) had a coat protein that differed by two amino acids, located between 209–210, from the other variants examined here (Figure 4). The Israel variant (Accession no. X15656) had a coat protein that differed by five amino acids, located between 213–217, from the other variants examined here (Figure 4). 

## 4. Discussion

The caveats surrounding previous reports of transovarial and/or sexual transmission of TYLCV variants by the B cryptic species have been explored in several laboratories, with variable and contradictory results. In this and other studies, transovarial and sexual transfer of TYLCV DNA has been reported to occur at differing frequencies. The frequencies ranged from zero to a very low percentage among the different whitefly instars to as high as 92% [16,30,31,34]. Several hypotheses have been advanced in support of transovarial- and mating-facilitated TYLCV transfer. However, the bases for the widely observed discrepancies among the different studies have remained unreconciled.

Studies designed to determine whether TYLCV transmission can occur through transovarial and/or sexual modes are based primarily on the ability to detect and/or quantitate virus DNA. Two relatively recent studies have shown interactions of the TYLCV coat protein with specific antibodies in F1 adults as well, potentially indicative of vertical transmission [26,34]. The accumulation of TYLCV DNA in F1 adults following transovarial transmission was at least one order of magnitude less than in whitefly adults that acquired the virus via phloem-feeding [26]. The accumulation of TYLCV DNA in initially (non-viruliferous) whitefly adults post-mating with viruliferous adults has not been quantitated. 

In this study, TYLCV DNA accumulation was documented in individual whitefly offspring (F1) following transovarial transfer and in mating partners. TYLCV DNA accumulation post transovarial and mating transfer was two to three orders of magnitude less than in individual whiteflies that acquired the virus by phloem-feeding. In addition, the inability to detect virions following transovarial and/or mating transfer by immunocapture PCR amplification, a sensitive assay, corroborates the assertion that even if there were sustained virus transcription, it was not sufficient to increase virion loads enough to lead to any epidemiological relevance. These results are consistent with those of the Bosco et al. study [33].

Accordingly, the observations reported in this study indicate that alternative, non-plant mediated modes of acquisition of the TYLCV Georgia variant by the whitefly vector did not result in whitefly-mediated transmission to tomato test plants. Indeed, TYLCV DNA was not detectable. None of the plants were positive for TYLCV DNA, and virus symptoms did not develop in tomato plants post-exposure to whitefly F1 adult offspring or to adult mating partners derived from transovarial and mating experiments, respectively, even when 26 to 50 adults were clip caged to each test plant and provided with an otherwise optimal IAP. These results have been corroborated by other studies that found no evidence for transovarial and/or sexual transmission of TYLCV, or subsequent transmission of the virus to tomato plants by offspring and/or mating partners [29,30,32,33]. Only two groups thus far have reported transovarial and/or sexual transmission of TYLCV to tomato plants [31,34,35]. In one study, the age of the female parent (F0 mother) was implicated as a major factor contributing to offspring-mediated transovarial transmission of TYLCV to the plant host [34]. This conclusion was based on evidence that only 10 F1 offspring of 11-day old viruliferous mothers (F0) (physiologically, when the vitellogenin content in females was most abundant), were capable of transmitting the virus to inoculated tomato plants, at a frequency of 33%. In contrast, the F1 offspring from one-day old viruliferous (F0) females failed to transmit TYCLV to tomato test plants despite an adequate IAP [34]. The same study showed that vitellogenin levels increased steadily from five to eleven days before precipitously dropping. Another study also detected TYLCV DNA in 57% of offspring (F1) but only with five-to-eight-day old mothers (F0 females) and contributed to 10% frequency of transmission to tomato plants [31]. Strikingly, in the current study, which provided F1 females of a comparable age (i.e., 6- to 13-days old post-eclosion) a 48 h IAP on non-infected tomato test plants did not result in any transmission of TYLCV. These results indicated that the age of the F0 mothers’ might not be the sole determinant of transovarial transfer of TYLCV DNA or virions in whiteflies. Also, the role of vitellogenin in the mating transfer of TYLCV DNA or virions between whiteflies, if any, is not known. Beyond the F1, transovarial transfer of TYLCV DNA also was observed in the F2 in this study. TYLCV DNA also has been documented to persist for at least two generations transovarially in earlier studies [31,34]. However, in those two studies, it is unclear whether TYLCV virions were also present, albeit TYLCV infection of test plants was reported. Nevertheless, detection of at least some level of TYLCV DNA molecules in two consecutive mating cohorts of whitefly may suggest that a low level of TYLCV DNA molecules that might be recalcitrant to whitefly defenses persist [43]. It also could be possible that defense responses such as autophagy past the germline barrier might not be as robust as in the hemolymph. Of course, this speculation needs to be examined and experimentally validated. TYLCV replication is an inconclusive phenomenon, the observed minimal transcription/replication exclusively in salivary glands [26], unlike other propagative viruses, does not explain the persistence of TYLCV DNA in presumably reproductive tissues for two consecutive generations.

Plant transmission of TYLCV (24 to 34%) following mating acquisition thus far has only been documented in one study [35]. In the current study, consistent with other reports, no TYLCV transmission to plants was observed following inoculation of plants with whiteflies that presumably acquired TYLCV by mating or transovarially. Although unequivocal, it is possible that the proposed age-modulated transovarial transmission with putative sexual transmission, and subsequent virus transmission by offspring to the plant host, are unique traits associated with a particular *B. tabaci* cryptic species haplotype and/or variant of TYLCV [34]. With over 400 known begomovirus species (ICTV) and several evaluated for non-feeding acquisition, the resultant plant transmission by any means other than phloem-feeding has only been documented for TYLCV and dolichos yellow mosaic virus [31,34,44]. Studies have indicated that non-feeding modes are highly unlikely to be relevant from an epidemiological standpoint to most or all begomovirus-*B. tabaci* cryptic species combinations among monopartite or bipartite viruses, including squash leaf curl virus, tobacco leaf curl virus, tomato leaf curl Sinaloa virus, TYLCSV, tomato yellow leaf curl China virus, papaya leaf curl China virus, cucurbit leaf crumple virus, and African cassava mosaic virus [32,33,45,46,47,48,49,50]. The results from these studies, and those reported here for TYLCV, strongly support the paradigm that the transmission of begomoviruses by the whitefly vector via non-feeding modes is not a conserved trait among begomoviruses but rather an uncommon occurrence even in the case of TYLCV. 

Transmissibility and the transmission specificity of begomoviruses by members of the *B. tabaci* cryptic species group are maintained by specific interactions of the virus capsid protein with mostly as yet ill-defined whitefly-encoded proteins [51,52]. Minor amino acid changes in the TYLCV capsid protein could render it incapable of transmission by whiteflies, while potentially retaining the capacity to infect plant hosts [53]. In the closely related TYLCSV, a group of amino acids located between residues 129 and 152 in the coat protein was essential for virus transmission [54]. Alignment of the coat protein amino acid sequences of selected TYLCV variants (GenBank database) with the coat protein of TYLCV Georgia variants revealed several variants with single amino acid substitutions, and only two variants differed by two and five amino acids—those from China and Israel, respectively (Figure 4). Of these, only two single amino acid substitutions in two Chinese variants were in the 129–152 region considered essential for transmission. Whether changes in the 129–152 region or in other regions of the coat protein could feasibly alter virus-vector interactions remains to be determined. 

Based on the low detection of TYLCV DNA observed for the B cryptic species in the experiments reported here, experiments designed to recapitulate transovarial and/or mating-associated transmission in the *B. tabaci* B cryptic species, the hypothesis that non-feeding modes of TYLCV transmission could provide an alternative explanation for how the virus might be maintained during the winter and/or summer months, when tomatoes are not cultivated in Georgia, can be discounted. The alternative hypothesis that virus over-winters/over-summers on feral hosts appears most likely. For example, the winter annual, henbit deadnettle, *Lamium amplexicaule* L., is reported as a TYLCV host in Korea [55]. This weed also occurs widely in southern Georgia, but preliminary testing indicated no TYLCV infection under field conditions (unpublished). Also, *Amaranthus retroflexus* L., a summer weed host in the southeastern United States, is a host of TYLCV [56]. More recently, another weed host, *Palmer amaranth*, *Amaranthus palmeri* S. Watson, in South Georgia in the United States has been identified as a TYLCV inoculum source and a whitefly reservoir [50]. Legarrea et al. (2020) demonstrated that TYLCV can be transmitted to *A. palmeri* from tomatoes and vice-versa, suggesting that this host and potentially others can be biologically relevant to annual reoccurrences of TYLCV epidemics in the southeastern United States [37]. Also, the role of the infection of tomato seedlings in greenhouses that are shipped between tomato-producing areas and the infection of volunteer tomato plants between the major growing seasons cannot be discounted as additional sources of inoculum that drive the reoccurring TYLCV epidemics in Georgia and in other southeastern states such as Florida in the United States [38].

## 5. Conclusions

Although some plant viruses that are persistent and/or propagative in their insect vector may be horizontally and vertically transferred to their offspring at low levels, their ensuing transmission to primary hosts through non-feeding transmission modes underscores their relevance to the infection cycle and recurring outbreaks. The primary mode of transmission of the monopartite begomovirus TYLCV is characterized as circulative and non-propagative [18,19,20,21,22], with transmission occurring only after virion translocation of the vector’s gut and salivary membrane barriers has been accomplished. TYLCV’s replication details within the vector are characterized by relatively mild and temporary replication exclusively in the salivary glands [26] in contrast to multiplication by many other propagative viruses (such as orthotospoviruses and rhabdoviruses) in multiple vector tissues [57,58]. This extensive replication of propagative viruses might facilitate germline invasion. By analogy, to facilitate non-feeding modes of transfer, germline and salivary membrane barriers must be breached. These accepted paradigms do not detract from the expectation that pathogens continuously explore evolutionary space, and in some instances could bypass such barriers [26]. Such a scenario could explain successful TYLCV transmission via non-feeding modes in some instances [31,34,35,44]. Overall, the results of our study indicated that for the Georgia variant of the Israel strain of TYLCV, virus DNA, but not virions, was detected at a very low frequency in the F1 offspring of viruliferous females and in females that mated with viruliferous males. Virions are necessary to initiate the infection cycle of begomoviruses in their plant hosts. As such, the inability to detect virions by immunocapture PCR amplification in F1 offspring following transovarial transfer and in mating partners following mating transfer suggests that transovarial- and/or mating transmission might not contribute to virus persistence in the TYLCV-*B. tabaci* pathosystem in Georgia, USA. Thus, our results show that non-feeding transmission modes are not epidemiologically relevant in the TYCLV-*B. tabaci* pathosystem, and alternate weed hosts or other plant inoculum sources support the annual reoccurrence of TYLCV. 

## Figures and Tables

**Figure 1 insects-15-00760-f001:**
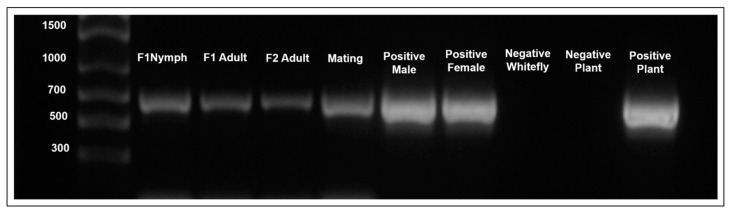
Gel photograph illustrating detection of tomato yellow leaf curl virus (TYLCV) DNA in nymphal offspring of viruliferous females that acquired TYLCV through feeding (F1 nymph), in adult offspring of viruliferous females that acquired TYLCV through feeding (F1 adult), in adult offspring of viruliferous females that acquired TYLCV DNA transovarially from F1 adults (F2 adult), in adult whiteflies that acquired TYLCV DNA via mating (mating), individual male and female adults that acquired the virus via feeding (positive male and female), adult whiteflies that fed on non-infected tomato plants (negative whitefly), non-infected tomato plant (negative plant), and TYLCV-infected tomato plant (positive plant). The total DNA was isolated and purified, amplified by PCR using the C2-1201 and C2-1800V2 primers, and analyzed by agarose gel (1%) electrophoresis. The expected size of the amplicon was 695 base pairs.

**Figure 2 insects-15-00760-f002:**
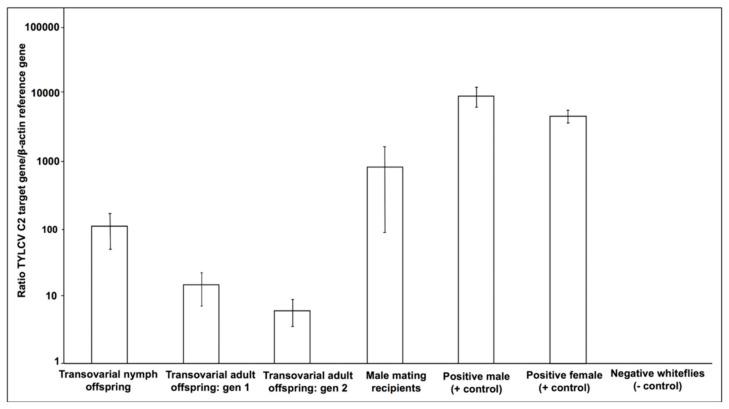
Bars ± standard errors indicate Georgia variant of tomato yellow leaf curl virus (TYLCV) accumulation based on quantitative PCR amplification. The DNA was isolated and purified from individual whitefly offspring and mated adults resulting from the transovarial and mating transfer experiments and subjected to quantitative PCR amplification. The relative TYLCV DNA accumulation is presented as the ratio of virus DNA to the internal baseline gene, ß-actin.

**Figure 3 insects-15-00760-f003:**
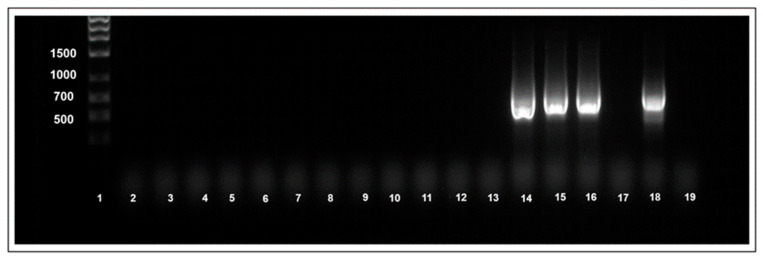
Gel electrophoresis photograph of immunocapture PCR of whitefly samples that tested positive in transovarial and mating transfer experiments. Individual insects that acquired the virus via feeding are included for comparison. Lane: 1Kb ladder; lanes 2 to 9: homogenates of whiteflies that tested positive for tomato yellow leaf curl virus (TYLCV) DNA in transovarial transfer experiment; lanes 10 to 13: homogenates of whiteflies that tested positive for TYLCV DNA in mating transfer experiment; lanes 14 to16: homogenates of whiteflies that tested positive for TYLCV DNA following feeding on TYLCV infected tomato; lane 17: non-infected whitefly; lane 18: TYLCV infected tomato (+ve control); and lane 19: non-infected tomato.

**Figure 4 insects-15-00760-f004:**
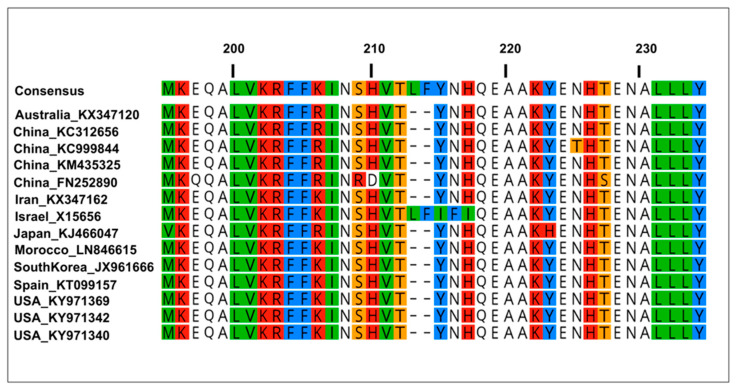
Comparison of partial coat protein amino acid sequences of Georgia tomato yellow leaf curl (TYLCV) variants in the southeastern United States with selected representative samples from worldwide.

**Table 1 insects-15-00760-t001:** List of the genome sequences for selected variants of tomato yellow leaf curl virus (TYLCV) and the respective GenBank accession numbers used for the alignment of coat protein amino acid sequences with the coat protein amino acid sequences for the TYLCV Georgia variants.

Genome	GenBank Accession Number
Australia28-06	KX347120
China113-12	KC312656
China137-12	KC999844
China160-13	KM435325
China7-07	FN252890
Iran23-09	KX347162
Israel1-89	X15656
Japan7-11	KJ466047
Morocco17-14	LN846615
SouthKorea23-12	JX961666
Spain4-11	KT099157
USAGeorgia15-15	KY971369
USAGeorgia28-16	KY971342
USAGeorgia34-16	KY971340

**Table 2 insects-15-00760-t002:** Results of transovarial transfer experiments, based on PCR detection of the Georgia variant of tomato yellow leaf curl virus (TYLCV) DNA in offspring resulting from viruliferous female parents.

Life Stage	Experimental Repeats	Number Positive/Total Number	Percent TYLCV-Positive
First-generation Nymphs (F1)	1	2/50	4 4
	2	2/50	
First-generation Adults (F1)	1	1/51	2 2
	2	1/51	
Second-generation adults (F2)	1	0/51	0 4
	2	2/52	

**Table 3 insects-15-00760-t003:** Results of sexual transfer (mating) experiments, based on PCR detection of the Georgia variant of tomato yellow leaf curl virus (TYLCV) DNA in adults resulting from reciprocal crosses between ‘initially’ non-viruliferous whiteflies and viruliferous whiteflies.

	Experimental Repeats	Whiteflies Positive for TYLCV DNA after Mating/Total Number Tested	Total	Percent TYLCV-Positive
Non-viruliferous females mated with viruliferous males	1	0/26	0/53	0
	2	0/27		
Non-viruliferous males mated with viruliferous females	1	1/26	2/52	4
	2	1/26		

**Table 4 insects-15-00760-t004:** Results of inoculation of tomato test plants by whitefly offspring resulting from transovarial and mating transfer experiments, based on PCR detection of the Georgia variant of tomato yellow leaf curl virus (TYLCV) DNA.

	Experimental Repeats	Number of TYLCV-Infected Plants/Total Number Tested	Total	Percent of TYLCV-Infected Plants
Adult offspring (F1)	1	0/10	0/20	0
	2	0/10		
Non-viruliferous females mated with viruliferous males	1	0/6	0/12	0
	2	0/6		
Non-viruliferous males mated with viruliferous females	1	0/6	0/12	0
	2	0/6		

**Table 5 insects-15-00760-t005:** Results of immunocapture-PCR detection of tomato yellow leaf curl virus (TYLCV) virions in whitefly offspring and mated adults resulting from the transovarial and mating experiments.

	Experimental Repeats	Number of Individuals Positive for TYLCV Virions/Total Number of Individuals	Total	Percent of Whiteflies with Virions
Adult offspring (F1)	1	0/49	0/99	0
	2	0/50		
Non-viruliferous females mated with viruliferous males	1	0/26	0/52	0
	2	0/26		
Non-viruliferous males mated with viruliferous females	1	0/26	0/53	0
	2	0/27		

## Data Availability

The data presented in this study are all included on the manuscript.

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
