# Peer review of "Non-Feeding Transmission Modes of the Tomato Yellow Leaf Curl Virus by the Whitefly Bemisia tabaci Do Not Contribute to Reoccurring Leaf Curl Outbreaks in Tomato"

_insects, 2024, doi:10.3390/insects15100760_

Round 1

Reviewer 1 Report

Comments and Suggestions for Authors

The non-feeding transmission of begomoviruses and particularly of Tomato yellow leaf curl virus (TYLCV) is a long-standing question that was addressed by several groups of scientists, and interestingly a report on the possibility of transovarial transmission of TYLCV disease (TYLCD), quoted in this manuscript, was published even before TYLCV was known to be a geminivirus (Iannou, 1985). The important number of reports on this topic is obviously related to the agronomic importance of the disease associated with TYLCV and its global spread which made it easily accessible to a large community of virologists all over the world. Then, with the increasing knowledge obtained on this economically important virus, TYLCV became a model geminivirus that became attractive also for more fundamental research. Hence, the question of non-feeding transmission of TYLCV was addressed by the scientific community with both fundamental and applied research objectives. In this study the authors address the question of non-feeding transmission of TYLCV with an applied objective. Indeed, they wanted to know if a potential non-feeding transmission of TYLCV may participate to the re-occurrence of TYLCV after non-tomato growing seasons in Georgia (USA).

Using various PCR techniques, they show that a TYLCV isolate from Georgia can be transmitted transovarially from viruliferous whiteflies to a first and second generation of progenies, and that it can be sexually transmitted from females to males but not vice versa. Using qPCR they showed that viral DNA amount in progenies of viruliferous whiteflies and in male mating recipients were 1 to 3 orders of magnitude lower than the amount of viral DNA in whiteflies that acquired TYLCV by feeding. The non-feeding transmitted virus was not detectable with Immuno-capture PCR. This last result seems to be consistent with the fact that whiteflies that were detected positive for TYLCV following transovarial or sexual processes were not able to transmit TYLCV to tomato plants. Comparisons of peptide sequences of TYLCV capsid protein were non-conclusive.

The objective of the study is clear and the technical approaches are adapted. The state of the art is conveniently summarized with most of the relevant papers (a few are missing, see below) and conflicting results are described regarding replication of TYLCV in its vector and its non-feeding transmission. The results are clearly presented and discussed in relation with the conflicting results. Based on their results the authors conclude that the non-feeding transmission of TYLCV does not seem to contribute to the re-occurrence of TYLCV after non-tomato seasons.

General comment:

According to the applied objective of the study, the non-feeding transmission of TYLCV DNA was considered negligible in relation to its epidemiological relevance. However, according to more fundamental objective, the fact that TYLCV DNA could be detectable in second generation progenies obtained in non-host plants from viruliferous whiteflies is quite fascinating. Hence, the stability of such TYLCV entities presumably without replication could be discussed in relation to their resistance to defence or degradation mechanisms of the whitefly.

Some specific comments:

Lines 31-34, not clear: 4 and 2% corresponding to three types of progenies, respectively? Rephrase?

Line 43: transovarial or sexual transmission of TYLCV DNA were detected in this study. So how can it be considered as probable and not proven?

Line 75: for the non-propagative mode the following reference should be added:

Sanchez-Campos, S., Rodriguez-Negrete, E.A., Cruzado, L., Grande-Perez, A., Bejarano, E.R., Navas-Castillo, J., Moriones, E., 2016. Tomato yellow leaf curl virus: No evidence for replication in the insect vector Bemisia tabaci. Sci Rep 6.

Line 75-77: It is stated "three studies" but only two are quoted. I suppose that the omitted one is the following of the reference list:

He, Y.Z., Wang, Y.M., Yin, T.Y., Fiallo-Oliv, E., Liu, Y.Q., Hanley-Bowdoin, L., Wang, X.W., 2020. A plant DNA virus replicates in the salivary glands of its insect vector via recruitment of host DNA synthesis machinery. Proceedings of the National Academy of Sciences of the United States of America 117, 16928-16937.

Line 77: Although CP is indispensable for vector transmission, are there any report of observations of begomovirus virions in Bemisia? If not, it should be stated that virions are presumed to circulate through the vector.

Line 81: For the receptor mediated endocytosis, the following reference should be added

Zhao et al. 2020: Zhao, J., Lei, T., Zhang, X.J., Yin, T.Y., Wang, X.W., Liu, S.S., 2020. A vector whitefly endocytic receptor facilitates the entry of begomoviruses into its midgut cells via binding to virion capsid proteins. Plos Pathogens 16.

Lines 151-158: As far as I understand the most critical contamination of PCR tests is not with the viruliferous females that laid eggs on cotton following AAP; they are by definition virus positive. PCR contaminations are more critical with the progenies that are produced on cotton following the laying period (lines 167-177). Was there any surface sterilization with these progeny whiteflies?

Line 256: what is the use of respectively in this sentence

Line 257: why n=50 and not 25 if it is per mating combination. Not dear

Lines 280-281: Please reword this sentence in which the word "respectively" is confusing. Moreover, in the case of the 2nd generation the 2 percent correspond to an average between the repeats.

Fig.1: As cotton is used as non-host plant, a test of a sample of cotton plants following the access of viruliferous whiteflies would have been useful.

Lines 458- 460: For most of the hundreds of begomoviruses, non-feeding transmission was not tested. Thus, it is not possible to base any strong conclusion on potential non-feeding transmission for them.  

Comments on the Quality of English Language

Some rephrasing may be needed (see specific comments)

Author Response

Dear Reviewer,

My co-authors and I greatly appreciate your comments. We believe that your comments have significantly improved our manuscript. We have carefully considered each comment and have tried to address them to the best of our abilities. Our explanation for each comment is included below. Our replies are in bold font. The revisions also are tracked on the revised version of the manuscript with a red font.

Reviewer 1
The non-feeding transmission of begomoviruses and particularly of Tomato yellow leaf curl virus (TYLCV) is a long-standing question that was addressed by several groups of scientists, and interestingly a report on the possibility of transovarial transmission of TYLCV disease (TYLCD), quoted in this manuscript, was published even before TYLCV was known to be a geminivirus (Iannou, 1985). The important number of reports on this topic is obviously related to the agronomic importance of the disease associated with TYLCV and its global spread which made it easily accessible to a large community of virologists all over the world. Then, with the increasing knowledge obtained on this economically important virus, TYLCV became a model geminivirus that became attractive also for more fundamental research. Hence, the question of non-feeding transmission of TYLCV was addressed by the scientific community with both fundamental and applied research objectives. In this study the authors address the question of non-feeding transmission of TYLCV with an applied objective. Indeed, they wanted to know if a potential non-feeding transmission of TYLCV may participate to the re-occurrence of TYLCV after non-tomato growing seasons in Georgia (USA). 

Using various PCR techniques, they show that a TYLCV isolate from Georgia can be transmitted transovarially from viruliferous whiteflies to a first and second generation of progenies, and that it can be sexually transmitted from females to males but not vice versa. Using qPCR they showed that viral DNA amount in progenies of viruliferous whiteflies and in male mating recipients were 1 to 3 orders of magnitude lower than the amount of viral DNA in whiteflies that acquired TYLCV by feeding. The non-feeding transmitted virus was not detectable with Immuno-capture PCR. This last result seems to be consistent with the fact that whiteflies that were detected positive for TYLCV following transovarial or sexual processes were not able to transmit TYLCV to tomato plants. Comparisons of peptide sequences of TYLCV capsid protein were non-conclusive. 

The objective of the study is clear and the technical approaches are adapted. The state of the art is conveniently summarized with most of the relevant papers (a few are missing, see below) and conflicting results are described regarding replication of TYLCV in its vector and its non-feeding transmission. The results are clearly presented and discussed in relation with the conflicting results. Based on their results the authors conclude that the non-feeding transmission of TYLCV does not seem to contribute to the re-occurrence of TYLCV after non-tomato seasons. 

General comment:

According to the applied objective of the study, the non-feeding transmission of TYLCV DNA was considered negligible in relation to its epidemiological relevance. However, according to more fundamental objective, the fact that TYLCV DNA could be detectable in second generation progenies obtained in non-host plants from viruliferous whiteflies is quite fascinating. Hence, the stability of such TYLCV entities presumably without replication could be discussed in relation to their resistance to defence or degradation mechanisms of the whitefly. 

TYLCV DNA also has been documented to persist for at least two generations transovarially in earlier studies [31,34]. However, in those two studies, it is unclear whether TYLCV virions were also present, albeit TYLCV infection of test plants was reported. Nevertheless, detection of at least some level of TYLCV DNA molecules in two consecutive mating cohorts of whitefly may suggest that a low level of TYLCV DNA molecules persist that might be recalcitrant to whitefly defenses [43]. It also could be possible that defense responses such as autophagy past the germ line barrier might not be as robust as in the hemolymph. Of course, this speculation needs to be examined and experimentally validated. TYLCV replication is an inconclusive phenomenon, unlike other propagative viruses, the observed minimal transcription/replication is exclusively in the salivary glands [26], and yet does not explain persistence of TYLCV DNA in presumably reproductive tissues for two consecutive generations. Some of this information is included in the discussion section in lines 483-494.

Some specific comments:

Lines 31-34, not clear: 4 and 2% corresponding to three types of progenies, respectively? Rephrase?

Has been rephrased as TYLCV DNA was detectable in four, two, and two percent of first generation fourth-instar nymphs, and first- and second-generation adults, respectively. Currently in lines 32-34.

Line 43: transovarial or sexual transmission of TYLCV DNA were detected in this study. So how can it be considered as probable and not proven?

Though possibilities may exist for transovarial or sexual transmission or transfer of TYLCV DNA at low frequencies, there is no proof that biologically active TYLCV virions have been transmitted in the studies in question. Virions are essential for subsequent plant infectivity. Prior research also supports our hypothesis and in fact, the canonical paradigm that virions are essential for vector-mediated transmission to occur (Bosco et al. 2004 with TYLCSV; cited in this manuscript). Revised version lines 40-45.

Line 75: for the non-propagative mode the following reference should be added:

Sanchez-Campos, S., Rodriguez-Negrete, E.A., Cruzado, L., Grande-Perez, A., Bejarano, E.R., Navas-Castillo, J., Moriones, E., 2016. Tomato yellow leaf curl virus: No evidence for replication in the insect vector Bemisia tabaci. Sci Rep 6.

 Thanks for providing us this reference, we appreciate it. This has now been included as reference 23.

Line 75-77: It is stated "three studies" but only two are quoted. I suppose that the omitted one is the following of the reference list:

He, Y.Z., Wang, Y.M., Yin, T.Y., Fiallo-Oliv, E., Liu, Y.Q., Hanley-Bowdoin, L., Wang, X.W., 2020. A plant DNA virus replicates in the salivary glands of its insect vector via recruitment of host DNA synthesis machinery. Proceedings of the National Academy of Sciences of the United States of America 117, 16928-16937.

Thanks again, this reference has now been included as well. This has now been included as reference 26.

Line 77: Although CP is indispensable for vector transmission, are there any report of observations of begomovirus virions in Bemisia? If not, it should be stated that virions are presumed to circulate through the vector. 

The information has been altered to suggest that only virions are presumed to circulate through the vector. Currently in lines 80-92.

Line 81: For the receptor mediated endocytosis, the following reference should be added

Zhao et al. 2020: Zhao, J., Lei, T., Zhang, X.J., Yin, T.Y., Wang, X.W., Liu, S.S., 2020. A vector whitefly endocytic receptor facilitates the entry of begomoviruses into its midgut cells via binding to virion capsid proteins. Plos Pathogens 16.

Thanks again, this reference has now been included as reference 28.

Lines 151-158: As far as I understand the most critical contamination of PCR tests is not with the viruliferous females that laid eggs on cotton following AAP; they are by definition virus positive. PCR contaminations are more critical with the progenies that are produced on cotton following the laying period (lines 167-177). Was there any surface sterilization with these progeny whiteflies?

Yes, the progenies were tested for transfer of TYLCV DNA following surface sterilization as well. Surface sterilization was performed for F1 nymphs and adults as well as for F2 adults.  The same protocol was undertaken for mating transmission.  This information is included in lines 170-172 and 185-186.

Line 256: what is the use of respectively in this sentence

This is now removed and appropriately edited.

Line 257: why n=50 and not 25 if it is per mating combination. Not dear

Should have been listed as n=25. The experiment included twenty-five replicates for each mating combination (25 mating pairs) and twenty-five for reciprocal combination (25 mating pairs). This experiment was conducted twice, bringing the number to 50 for each mating combination. This has now been clarified. This information is included in lines 280-281.  

Lines 280-281: Please reword this sentence in which the word "respectively" is confusing. Moreover, in the case of the 2nd generation the 2 percent correspond to an average between the repeats. 

The sentence has been rephrased by removing the word ‘respectively’.

Fig.1: As cotton is used as non-host plant, a test of a sample of cotton plants following the access of viruliferous whiteflies would have been useful. 

In hindsight, yes. Cotton has been reported as a non-host for TYLCV in several studies. If cotton plants were infected with TYLCV following F0 adults feeding in our study, we would have expected a much higher TYLCV infection frequency in F1 and F2 offspring. Also, virions in F1 and F2 whiteflies would have been detected by immunocapture PCR, and we would have observed plant transmissibility in transovarial-plant transmission assays.

Lines 458- 460: For most of the hundreds of begomoviruses, non-feeding transmission was not tested. Thus, it is not possible to base any strong conclusion on potential non-feeding transmission for them.  

We agree. This information has now been revised and included in lines 503-506.

Yours sincerely,

Rajagopalbabu Srinivasan

Reviewer 2 Report

Comments and Suggestions for Authors

Insects

Non-Feeding transmission modes of tomato yellow leaf curl virus by the whitefly Bemisia tabaci B cryptic species might not contribute to reoccurring leaf curl outbreaks in tomato

Marchant et al.

Review Comments

This manuscript addresses the potential for transovarial transmission and transsexual transmission of TYLCV by whitefly vectors, a topic that has been explored in several previous studies but remains inconclusive. While this study employs similar transmission assays used in earlier research and lack novelty, the results of this study contribute additional evidence to support the conclusion that non-feeding transmission does not occur for TYLCV. Although this study used Georgia variant of TYLCV, the comparison of coat protein amino acid sequence with other TYLCV strains showed no significant difference.

However, the manuscript requires significant revision in terms of writing. The materials and methods section is particularly poorly written with many important details are missing. In addition, the manuscript needs through proofreading for punctuation errors, such as the missing period on line 65 and missing right parentheses on line 161.

Comments in details:

Line 3 (Title): The manuscript refers to Bemisia tabaci B cryptic species. It was previously known as the “B biotype,” but the preferred name now is "Middle East-Asia Minor 1 (MEAM1)." I recommend using this standard name to avoid introducing another name for the species.

Line 17: change “non-feeding modes” to “non-feeding transmission modes”

Line 20: add “virus” before transmission assays

Line 31-32: The phrase “TYLCV DNA was detectable in…” should specify the treatment that led to this detection. The following sentence of the manuscript provides a good example of how to clearly explain the treatment.

Line 38: Explain what immunocapture PCR is and clarify why multiple detection methods were necessary in the study in addition to end-point PCR assays.

Line 75-77: only two references (21,22) are cited for a sentence that mentions three studies reported for TYLCV being replicated with the vector.

Line 97-97, 460-165: Consider adding another reference of Tomato Yellow Leaf Curl Thailand Virus regarding transovarial transmission by Li et al., 2021. https://doi.org/10.3390/insects12020181

Line 111: With these previous studies, what conclusion led to this study? what knowledge gap needs to be filled, and why is it important to understand non-feeding transmission in terms of virus epidemiology and management?

Line 112-121: Why is it important to test TYLCV Georgia variant? how does it differ from other variants? what is the economic impact of TYLCV in area? More information would strengthen the rational for the study.  

Line 114: Clarify what is meant by “in the southeastern United States.” Does this refer to the region where the B cryptic species was collected, or was the experiment conducted in this region?

Line 115-118: The explanation provided is not very convincing. First, other host plants besides tomato can also harbor TYLCV, potentially explaining the virus's overwintering during the tomato-free season. What is the host range, and have any non-crop hosts been identified and tested for TYLCV in or near tomato production areas? Second, for the virus to persist in vector populations without additional virus acquisition, it is crucial to determine whether the virus replicates within the vector, as well as investigating transovarial or transsexual transmission. Without virus replication, virus titer and infection rates in vector populations would decrease over time, even with non-feeding transmissions. However, TYLCV replication was not examined in this study, despite its importance to TYLCV epidemiology, and results from previous studies have been inconclusive.

Line 127-131: Is there a specific reason why the whiteflies collected from cotton and not tomato? Just curious whether there is any difference between the colonies reared on cotton versus tomato. Could switching host from cotton to tomato affect feeding behavior and virus acquisition rate?

Line 131-134: The purpose of this sentence is unclear and needs clarification.

Line 148-151: How were 3 days old adults obtained? Did the transfer from tomato to cotton occur immediately after the 3 days AAP? how was sex of whiteflies determined? how many females were transferred to cotton for oviposition?

Line 151: how were the females removed from the cotton plant?

Line 152-153: Does it mean Individual “whiteflies” or “tomato”? if it refers to whitefly, was the tomato plant surface-sterilized to remove honeydew residue on the leaves?

Line 174: how many whiteflies were used?

Line 196-211: Were positive controls using the original viruliferous whiteflies included?

Line 216: Does “After one week” refer to the end of the oviposition period? This need clarification.

Line 216-218: what was the result of this PCR test? Since this test verifies the infection status of the viruliferous parents, the results should be presented in the same section to justify the methods used. (Also, I did not notice it was presented in the results)

Line 220-222: how were the F1 whiteflies removed after the 48 h IAP?

Line 226: how many plants were inoculated per replication?

Line 230-232: Were 26 whiteflies used to inoculate one tomato plant, or how many whiteflies were used per tomato plant?

Ling 249-252: More details are needed here. For example, how much homogenates and antibody were used for the immunocapture? how the immunocapture were performed? using microtiter plate or magnetic beads? and what were the next steps before the PCR assays? and what was used as DNA template in the PCR? These details are missing in the next paragraph as well.

Line 262-267: How were the three coat protein sequences obtained? what was the source of the viruses? collected location, host plant, time?

Comments on the Quality of English Language

The manuscript requires significant revision in terms of writing (scientific writing, not English problem). The materials and methods section is particularly poorly written with many important details are missing. In addition, the manuscript needs through proofreading for punctuation errors, such as the missing period on line 65 and missing right parentheses on line 161.

Author Response

Dear Reviewer,

My co-authors and I greatly appreciate your comments. We believe that your comments have significantly improved our manuscript. We have carefully considered each comment and have tried to address them to the best of our abilities. Our explanation for each comment is included below. Our replies are in bold font. The revisions also are tracked on the revised version of the manuscript with a red font.

Review Comments

This manuscript addresses the potential for transovarial transmission and transsexual transmission of TYLCV by whitefly vectors, a topic that has been explored in several previous studies but remains inconclusive. While this study employs similar transmission assays used in earlier research and lack novelty, the results of this study contribute additional evidence to support the conclusion that non-feeding transmission does not occur for TYLCV. Although this study used Georgia variant of TYLCV, the comparison of coat protein amino acid sequence with other TYLCV strains showed no significant difference.

However, the manuscript requires significant revision in terms of writing. The materials and methods section is particularly poorly written with many important details are missing. In addition, the manuscript needs through proofreading for punctuation errors, such as the missing period on line 65 and missing right parentheses on line 161.

Comments in details:

Line 3 (Title): The manuscript refers to Bemisia tabaci B cryptic species. It was previously known as the “B biotype,” but the preferred name now is "Middle East-Asia Minor 1 (MEAM1)." I recommend using this standard name to avoid introducing another name for the species.

The nomenclature of Bemisia tabaci used in this manuscript is based on phylogeographical relationships established based on nuclear gene orthologs in addition to COI sequencing, with the former being the first evidence at the genome level for at least six cryptic species.  The new nomenclature refers to the NAFME (B) cryptic species, and it is one of six recognized (thus far) cryptic species. The findings are outlined in publications from Brown lab group and are highlighted in the recent review Brown et al. 2023. However, the point about acknowledging the previously well-recognized ‘MEAM 1’ has been addressed in lines 64-76, and to avoid confusion, cryptic species was removed from the title.

Line 17: change “non-feeding modes” to “non-feeding transmission modes”

Modified as suggested. Currently in line 17-18.

Line 20: add “virus” before transmission assays

Modified as suggested.

Line 31-32: The phrase “TYLCV DNA was detectable in…” should specify the treatment that led to this detection. The following sentence of the manuscript provides a good example of how to clearly explain the treatment.

Has now been specified in line 32-33.

Line 38: Explain what immunocapture PCR is and clarify why multiple detection methods were necessary in the study in addition to end-point PCR assays.

An explanation for why immunocapture PCR was used is succinctly explained in the abstract in lines 38-39, and additional details are included in the methods section as well in lines 269-277.

Line 75-77: only two references (21,22) are cited for a sentence that mentions three studies reported for TYLCV being replicated with the vector.

The additional reference (He et al. ) has now been added in line 81-83.

Line 97-97, 460-165: Consider adding another reference of Tomato Yellow Leaf Curl Thailand Virus regarding transovarial transmission by Li et al., 2021. https://doi.org/10.3390/insects12020181

The reference (now #28) has been added in both instances. We appreciate you for suggesting this reference. Thanks.

Line 111: With these previous studies, what conclusion led to this study? what knowledge gap needs to be filled, and why is it important to understand non-feeding transmission in terms of virus epidemiology and management?

Only a few studies have addressed the conundrum of mating-mediated begomovirus transmission, since the first reports were published claiming such non-canonical transmission modes for TYLCV. However, given the possibility that these modes may occur rarely and therefore could be significant, we investigated the possibilities to substantiate or refute the existing conundrum and related the results to biologically-relevant transmission associated with the epidemiology of the disease in Georgia B. tabaci populations. With mild winters, whitefly biology slows down but immatures in particular and sometimes adults can survive winter temperatures as long as the plant host does not freeze (Brown, personal observations in Arizona). Mating and transovarial transfer of biologically active TYLCV virions could feasibly facilitate TYLCV spread within a population and increase their inoculation potential. Currently in lines 117-123.

Line 112-121: Why is it important to test TYLCV Georgia variant? how does it differ from other variants? what is the economic impact of TYLCV in area? More information would strengthen the rational for the study.  

This information has now been edited in lines 124-126. Economic impact is explained in lines 60-61.

Line 114: Clarify what is meant by “in the southeastern United States.” Does this refer to the region where the B cryptic species was collected, or was the experiment conducted in this region?

This has been now explained in lines in the current version of the manuscript 128-135.

Line 115-118: The explanation provided is not very convincing. First, other host plants besides tomato can also harbor TYLCV, potentially explaining the virus's overwintering during the tomato-free season. What is the host range, and have any non-crop hosts been identified and tested for TYLCV in or near tomato production areas? Second, for the virus to persist in vector populations without additional virus acquisition, it is crucial to determine whether the virus replicates within the vector, as well as investigating transovarial or transsexual transmission. Without virus replication, virus titer and infection rates in vector populations would decrease over time, even with non-feeding transmissions. However, TYLCV replication was not examined in this study, despite its importance to TYLCV epidemiology, and results from previous studies have been inconclusive.

Other hosts of TYLCV have been examined such as crop hosts including eggplant and peppers, ornamentals such as petunia, and weeds such as amaranthus within the Southeastern United States [37,38,55,56].  Though almost all can be infected with TYLCV, viruliferous B. tabaci was only able to back transmit TYLCV from amaranthus to tomato. Numerous other weed hosts evaluated did not result in TYLCV infection and/or back transmission to tomato. Despite the relatively narrow host range, TYLCV incidences have become chronic in Georgia even with tomato-free seasons. This led to the speculation that non-feeding transmission of TYLCV could aid in sustaining inoculum within whiteflies. Nevertheless, the role of greenhouse infection in tomato seedlings that are shipped and volunteer tomato between seasons cannot be discounted as important during over-seasoning. Comments were added to the text, in lines 128-135 and in 544-549.

According to He et al [26], TYLCV replication was exclusively detected in the salivary glands is minimal and temporary. TYLCV accumulation levels in the gut are probably essential for TO and putative mating transfer/transmission if indeed they occur. 

Line 127-131: Is there a specific reason why the whiteflies collected from cotton and not tomato? Just curious whether there is any difference between the colonies reared on cotton versus tomato. Could switching host from cotton to tomato affect feeding behavior and virus acquisition rate?

Cotton is a non-host of TYLCV and has been used in our laboratory for the past 15 years (or so) and have had no trouble in getting B. tabaci to move to tomato. Also, cotton provides a platform to examine viruliferous whiteflies without the continued impact of host plant. We regularly maintain B. tabaci on cotton, and in fact it was originally collected on cotton.

Line 131-134: The purpose of this sentence is unclear and needs clarification.

This information is currently edited

Line 148-151: How were 3 days old adults obtained? Did the transfer from tomato to cotton occur immediately after the 3 days AAP? how was sex of whiteflies determined? how many females were transferred to cotton for oviposition?

Whiteflies are usually reared in cohorts with adults provided with an oviposition period by and removed. The adult insects emerging were checked every three days and retrieved by aspiration. After retrieving, they were clip caged to TYLCV-infected tomato plants and provided with an AAP of three days. Sex of the whiteflies was determined by immobilizing whiteflies on chilled blue ice and examination under a dissecting microscope at 10x. This Information currently in lines 160-168. Sex determination in whiteflies in lines 197-198.

Line 151: how were the females removed from the cotton plant?

They were removed by mechanical aspiration. Details in lines 160-161.

Line 152-153: Does it mean Individual “whiteflies” or “tomato”? if it refers to whitefly, was the tomato plant surface-sterilized to remove honeydew residue on the leaves?

It refers to individual whiteflies, and each whitefly was surface sterilized to avoid external contamination. Details in line 168-169.

Line 174: how many whiteflies were used?

Fifty whiteflies were used in each experiment, and the experiment was conducted twice. Overall, 100 individuals were used for each treatment.  Lines 189-193.

Line 196-211: Were positive controls using the original viruliferous whiteflies included?

The ‘initially-viruliferous’ whiteflies were tested with PCR amplification to determine if they harbored TYLCV DNA. Also, the initially non-viruliferous whiteflies were assayed by PCR to determine if TYLCV DNA was present in whiteflies following mating. Whiteflies exposed to TYLCV-infected tomato source plants for a 48 h AAP were assayed by PCR as positive controls. Details in lines 204-210.

Line 216: Does “After one week” refer to the end of the oviposition period? This need clarification.

Not necessarily the end of oviposition period. They were only exposed for one week. The adults were ~two weeks old at the end of exposure time. Details in lines 233-234.

Line 216-218: what was the result of this PCR test? Since this test verifies the infection status of the viruliferous parents, the results should be presented in the same section to justify the methods used. (Also, I did not notice it was presented in the results)

All the parents from the sub-cohort that acquired TYLCV by feeding tested positive for TYLCV. This information is included in lines 370-371.

Line 220-222: how were the F1 whiteflies removed after the 48 h IAP?

The whiteflies were removed along with the clip cage by gentle tapping, aspirator was occasionally used if need be. This information is included in lines 240-241.

Line 226: how many plants were inoculated per replication?

Two replicated experiments were carried out with ten plants each.  This information is included in lines 245-246.

Line 230-232: Were 26 whiteflies used to inoculate one tomato plant, or how many whiteflies were used per tomato plant?

That is correct, 26 per plant were used. Six plants were used for first experiment. The experiment was conducted two times in lines 251-258.

Ling 249-252: More details are needed here. For example, how much homogenates and antibody were used for the immunocapture? how the immunocapture were performed? using microtiter plate or magnetic beads? and what were the next steps before the PCR assays? and what was used as DNA template in the PCR? These details are missing in the next paragraph as well.

Immunocapture PCR was performed on 50 whitefly offspring homogenates (individually) using the polyclonal antibody raised against the TYLCV coat protein (Bioreba, Ebringen, Germany) (Ghanim and Czosnek 2000). The PCR tubes were coated with TYLCV antiserum ~100 μl at a 1:1000 dilution. The coated tubes were incubated for 3 h at 37°C and washed with 100 μl of washing buffer three times. The washed tubes were then incubated with 100 μl of individual whiteflies for 18 h at 4°C. Following the incubation period, PCR amplification was conducted as described above for transovarial and mating experiments using the primer pairs C2-1201 and C2-1800V2. Whiteflies allowed a 48 h AAP on TYLCV infected tomato plants were used as PCR positive control. These details are currently included in lines 269-278.

Line 262-267: How were the three coat protein sequences obtained? what was the source of the viruses? collected location, host plant, time?

The Georgia coat protein sequences used in this study were all obtained from whole genome sequencing of TYLCV variants from tomato foliar samples collected in 2015 and 2016. Details in lines 288-291.

Yours sincerely,

Rajagopalbabu Srinivasan

Reviewer 3 Report

Comments and Suggestions for Authors

This paper deals with transovarial and sexual transmission of the begomovirus TYLCV by the whitefly Bemisia tabaci. This topic has been studied before and is an object of debate since many years. The authors present results from transmission experiments as well as detections of the Georgian variant of the virus in a specific cryptic species (B) of B. tabaci. Whereas they detected the virus in adult and nymphal stages that developed from eggs of viruliferous insects – suggestive of a transovarial transmission of the virus – the numbers of positive cases were very few (0-4 %).  Feeding of these adults, as well as those obtained form eggs of B. tabaci couples investigating sexual transmission of the virus, failed to infect healthy tomato plants.

The research is straight forward, well executed with the appropriate controls, and well documented, The results support the conclusions of the paper.

The introduction and discussion section is also well documented.

Author Response

Dear Reviewer,

My co-authors and I greatly appreciate your comments. Thank you.

Reviewer 3. This paper deals with transovarial and sexual transmission of the begomovirus TYLCV by the whitefly Bemisia tabaci. This topic has been studied before and is an object of debate since many years. The authors present results from transmission experiments as well as detections of the Georgian variant of the virus in a specific cryptic species (B) of B. tabaci. Whereas they detected the virus in adult and nymphal stages that developed from eggs of viruliferous insects – suggestive of a transovarial transmission of the virus – the numbers of positive cases were very few (0-4 %).  Feeding of these adults, as well as those obtained form eggs of B. tabaci couples investigating sexual transmission of the virus, failed to infect healthy tomato plants. 

The research is straight forward, well executed with the appropriate controls, and well documented, The results support the conclusions of the paper.

The introduction and discussion section is also well documented.

Yours sincerely,

Rajagopalbabu Srinivasan

Round 2

Reviewer 2 Report

Comments and Suggestions for Authors

The authors have addressed all questions and suggestions and made revisions accordingly. The revised manuscript has detailed background information and clear materials and methods. I appreciate the authors' efforts. 

Author Response

Dear Reviewer, 

Thanks for your kind comments and for talking the time to review our manuscript. Your review certainly helped us improve the manuscript. We appreciate it. 

Best regards. 

Rajagopalbabu Srinivasan